# Therapeutic Potential of *Peucedanum japonicum* Thunb. and Its Active Components in a Delayed Corneal Wound Healing Model Following Blue Light Irradiation-Induced Oxidative Stress

**DOI:** 10.3390/antiox12061171

**Published:** 2023-05-29

**Authors:** Wan Seok Kang, Eun Kim, Hakjoon Choi, Ki Hoon Lee, Kyeong Jo Kim, Dosung Lim, Su-young Choi, Youngbae Kim, Seon ah Son, Jin Seok Kim, Sunoh Kim

**Affiliations:** Central R&D Center, B&Tech Co., Ltd., Naju 58205, Republic of Korea; kws2602@hanmail.net (W.S.K.); rubsang84@gmail.com (E.K.); ohchj12@naver.com (H.C.); leekh3261@daum.net (K.H.L.); kkkjzzang@nate.com (K.J.K.); lds9509@gmail.com (D.L.); csy971016@naver.com (S.-y.C.); unkr2003@naver.com (Y.K.); suna7856@nate.com (S.a.S.); keki2000@naver.com (J.S.K.)

**Keywords:** blue light, corneal wound healing, reactive oxygen species (ROS), *Peucedanum japonicum* Thunb

## Abstract

Blue light is reported to be harmful to eyes by inducing reactive oxygen species (ROS). Herein, the roles of *Peucedanum japonicum* Thunb. leaf extract (PJE) in corneal wound healing under blue light irradiation are investigated. Blue-light-irradiated human corneal epithelial cells (HCECs) show increased intracellular ROS levels and delayed wound healing without a change in survival, and these effects are reversed by PJE treatment. In acute toxicity tests, a single oral administration of PJE (5000 mg/kg) does not induce any signs of clinical toxicity or body weight changes for 15 days post-administration. Rats with OD (oculus dexter, right eye) corneal wounds are divided into seven treatment groups: NL (nonwounded OS (oculus sinister, left eye)), NR (wounded OD), BL (wounded OD + blue light (BL)), and PJE (BL + 25, 50, 100, 200 mg/kg). Blue-light-induced delayed wound healing is dose-dependently recovered by orally administering PJE once daily starting 5 days before wound generation. The reduced tear volume in both eyes in the BL group is also restored by PJE. Forty-eight hours after wound generation, the numbers of inflammatory and apoptotic cells and the expression levels of interleukin-6 (IL-6) largely increase in the BL group, but these values return to almost normal after PJE treatment. The key components of PJE, identified by high-performance liquid chromatography (HPLC) fractionation, are CA, neochlorogenic acid (NCA), and cryptochlorogenic acid (CCA). Each CA isomer effectively reverses the delayed wound healing and excessive ROS production, and their mixture synergistically enhances these effects. The expression of messenger RNAs (mRNAs) related to ROS, such as SOD1, CAT, GPX1, GSTM1, GSTP1, HO-1, and TRXR1, is significantly upregulated by PJE, its components, and the component mixture. Therefore, PJE protects against blue-light-induced delayed corneal wound healing via its antioxidative, anti-inflammatory, and antiapoptotic effects mechanistically related to ROS production.

## 1. Introduction

People are currently exposed to various lights from household appliances such as mobile phones, TVs, personal computers, and indoor illuminations of outdoor lighting, such as neon signs and decorative lighting; moreover, certain professionals are exposed to special types of light. The development of artificial lights, especially light-emitting diodes (LEDs), has changed many aspects of the modern lifestyle. LEDs have many advantages over incandescent lamps, such as a long lifetime, energy efficiency, easy control, low thermogenic, and good durability and stability. These features have promoted the development of the light industry, and we now use and enjoy LEDs every day [1]. In particular, the invention of bright blue LEDs has led to the production of a new form of high-efficiency white light and the ability to display of a full range of colors more clearly [2]. However, increasing artificial light exposure causes eyes to become more tired, even though human eyes have evolved to be suitable for living under sunlight [3]. LEDs emit narrow-spectrum light that is different from that emitted from incandescent lamps or naturally. White light is generated by combining two LEDs emitting shorter and longer wavelengths or more than three LEDs emitting red, green, and blue light; thus, blue LEDs are essential [4]. However, blue LEDs emit at a short wavelength, approximately 400 nm to 480 nm, with a peak at approximately 450 nm, which is in the most harmful range of visible light because of its high energy [5]. Recent studies have experimentally examined the harmful effects of blue light by investigating entire eye regions, from the cornea to the retina.

The cornea is the first eye structure that is exposed to the environment and functions as the first barrier, as well as a window for clear sight, but it is vulnerable to environmental risk factors such as air pollution, irradiation, dryness, and abrasion. Blue light induces reactive oxygen species (ROS) production, inflammation, and apoptosis in human corneal epithelial cells (HCECs) and animal models [6,7,8]. Oxidative stress activates NOD, LRR, and the pyrin domain-containing protein 3 (NLRP3) inflammasome and increases the secretion of both interleukin-1 (IL-1) and interleukin-6 (IL-6), which leads to inflammatory cell recruitment and tear film instability and causes hyperosmotic conditions on the eye surface. It has also been reported that natural extracts containing antioxidants can protect the eyes from these harmful effects of blue light [6]. A recent study reported that Sprague–Dawley (SD) rats irradiated with 100 lux of blue light for 12 h a day for 28 days showed tear instability, an increase in inflammatory factors, and histological degeneration, including a decrease in the number of microvilli in epithelial cells, all of which are characteristics similar to those observed with dry eye disease [9]. Damage from blue light occurs not only in the cornea but also in the lens and retina, and oxidative stress is still a key mechanism of injury to both of these eye structures. The lens absorbs blue light to block retinal damage, but continuous exposure to blue light increases oxidative stress in human lens epithelial cells (hLECs) and causes severe DNA damage, which is associated with the development of cataracts [10]. In the retina, blue light passing through the lens not only causes oxidative stress but also activates inflammatory factors (IL-1β, caspase-1, and NLRP3), microglia, and apoptosis, resulting in degenerative structural changes and retinopathy [11,12,13].

Although many studies have presented the above experimental data, the hazards of blue light irradiation under everyday conditions remain controversial. A report calculated the blue-light-weighted radiances of computer monitors, laptop screens, tablets, and smartphones and concluded that blue light emitted from these manufactured light sources does not exceed the hazard limits even after long-term use [14]. However, cumulative data acquired under various illumination conditions, including illuminance levels, exposure time, and number of cycles, demonstrated that the impacts of blue light cannot be ignored. Recent studies have suggested that antioxidants such as lutein, anthocyanin, curcumin, and vitamin E protect against light phototoxicity, which is induced mainly by oxidative stress [15,16,17]. However, although many therapeutic candidates have been suggested, actual remedies for corneal injury have not been sufficiently investigated.

Corneal abrasion commonly occurs from daily activities, such as exposure to dust and chemicals, surgery, and accidents [18,19]. Shallow scratches on the eye surface rapidly recover within 1~2 days because of epithelial cell proliferation and migration under normal conditions, but corneal recovery can be delayed after severe wounding or under metabolic disorder conditions [20,21]. Furthermore, there are no therapeutics suitable for corneal wound healing under these circumstances. Unfortunately, most studies have focused on the effects of blue light on oxidative stress, and its effects on corneal epithelial cell wound healing are rarely investigated. A recent study showed that low-energy blue light exposure delayed wound healing and increased ROS levels in HCECs [22]. Therefore, this study was designed to explore the effects of blue light on delayed corneal wound healing in vitro and in vivo and to investigate the protective mechanisms of a potential therapeutic reagent discovered in our previous study.

*Peucedanum japonicum* Thunb. (family *Apiaceae*) has been traditionally consumed not only as a vegetable in the East Asia region but also as a medicine to treat colds, rheumatoid arthritis, headaches, and inflammatory diseases [23]. *P. japonicum* contains coumarins, inositols, chromones, phenolic compounds, polyacetylenes, and steroid glycosides [24,25]. The extract of this plant has been reported to have various beneficial effects on obesity, allergies, oxidative stress, and inflammation [26,27,28,29,30]. In particular, our previous study confirmed that the leaf extract of *P. japonicum* (PJE) protected against damage to corneal epithelial cells induced by urban particulate matter (UPM) exposure via exerting antioxidative, anti-inflammatory, and antiapoptosis effects and strongly enhancing corneal wound healing in in vitro and in vivo models [31]. Chlorogenic acid (CA), neochlorogenic acid (NCA), and cryptochlorogenic acid (CCA) are known to be major constituents of PJE, and treatment with the mixture of these three compounds showed the maximum effect on HCECs in the scratch wound healing assay, even though treatment with each compound alone was also effective.

Therefore, PJE application with blue light exposure was assessed for its protective effects on corneal wound healing, and the suggested mechanisms and distinct characteristics of each active compound were identified.

## 2. Materials and Methods

### 2.1. Blue Light Irradiation Device

The blue backlight (L-LIGHT, Cheonan, Republic of Korea) used for the in vitro experiments was 100 × 150 mm in size and consisted of 288 blue LED chips emitting at a peak wavelength of 465 nm through an acryl diffuser (Figure 1A). The cell plate was placed directly above the LED backlight to receive irradiation in a CO_2_ incubator. A stainless-steel cabinet for housing animal cages was designed by our group for this study and was made by IWOO Scientific Corporation (Seoul, Republic of Korea) (Figure 1B). The cabinet was 1200 × 550 × 620 mm in size and divided into six compartments with a stainless wall for the animal cages. Each compartment was equipped with six lines of LEDs in the left, right, top, and back sidewalls, and each line contained 27 or 21 LED chips (on the back side only) emitting at a peak wavelength of 465 nm; thus, each wall on the left, right, and top sides had 162 LED chips, and the back sidewall had 126 LED chips. A fan was included on the back side of each compartment for ventilation. A portable acryl diffuser was also placed in each compartment, and an animal cage was inserted into the inner space of the diffuser. Holes (40 × 80 mm) spaced 20 mm apart were made in the cabinet doors for air flow. Blue light illumination was measured by a luxmeter (Uyigao, Shenzhen, China) before every experiment and adjusted with a controller.

### 2.2. Sample Preparation and High-Performance Liquid Chromatography (HPLC) Analysis

PJE was prepared as described in our previous study [31]. Briefly, the dried leaves of *P. japonicum* Thunb. were extracted using 20 volumes of water at 100 °C for 4 h. The spray-dried powder was taken as PJE. It was fractionated through solvents 𝑛-hexane, chloroform, ethyl acetate, and 𝑛-butanol and the remaining fraction was used as the water fraction. HPLC analysis was carried out with an Agilent 1260 HPLC system (Agilent Technologies, Palo Alto, CA, USA) equipped with an Eclipse XDB-C18 column (4.6 × 250 mm, 5 μm, Agilent Technologies).

### 2.3. Human Corneal Epithelial Cell (HCEC) Culture

The immortalized HCECs was purchased from RIKEN BioSource Center (Tokyo, Japan). The cell was cultured in Dulbecco’s modified Eagle’s medium/Ham’s F-12 (DMEM/F-12) (Welgene, Daegu, Republic of Korea), adding 5% fetal bovine serum, 10 ng/mL epidermal growth factor, 5 μg/mL insulin, and 0.5% dimethyl sulfoxide (DMSO).

### 2.4. Scratch Wound Healing Assay

The migration activity of HCECs was measured through scratch wound healing assay to assess the effect of blue light on cell migration function. The scratch wound healing assay was performed as previously reported [31]. Briefly, the cells were preincubated with PJE (10–300 μg/mL), fractions of PJE (1–30 μg/mL), or chlorogenic acid (CA, Sigma–Aldrich, St. Louis, MO, USA) and its isomers (neochlorogenic acid (NCA) (TCI, Tokyo, Japan), and cryptochlorogenic acid (CCA) (Cayman Chemicals, MI, USA)) (0.028–0.846 μM) suspended in growth media for 24 h. A steady scratch was generated using a sterile 200 μL pipette tip through crossing the cells straightly. The media was changed to fresh media with the same treatment conditions, and the images were taken using a microscope equipped with a digital camera (Leica Microsystems, Wetzlar, Germany) and used as 0 h. Then, the cells were irradiated with blue light (1200 lux) by placing the plate on the blue LED emitting a peak wavelength of 465 nm for 8 h and photographs were taken. The migration rate was calculated by determining the closed area relative to the area at 0 h; measurements were performed by ImageJ software (National Institutes of Health, Bethesda, MD, USA). The experiment was repeated three times, and the area calculation of each experiment was measured at least more than three times.

### 2.5. Cell Viability

Cell viability was measured as previous reported [31]. Briefly, cells (2 × 10^4^ cells/well) were seeded in 96-well plates for 24 h, and various concentrations of PJE (10–300 μg/mL), fractions of PJE (1–30 μg/mL) or CAs (0.028–0.846 μM) were added for 24 h. Then, the cells were exposed to blue light by placing the plate under a blue LED for 8 h or 30 min. The cells incubated with MTT solution and measured absorbance as previously described.

### 2.6. ROS Assay

Intracellular ROS levels were assessed by 2′,7′-dichlorofluorescein diacetate (DCFH-DA) (Sigma–Aldrich, St. Louis, MO, USA) as previously reported [32]. Briefly, the cells were incubated with 2.5 μM DCFH-DA for 30 min and washed with cold PBS twice. The fluorescence at 485 nm excitation and 535 nm emission wavelength was measured using a Tecan multimode microplate reader (Tecan Trading AG, Männedorf, Switzerland).

### 2.7. Real-Time Polymerase Chain Reaction (PCR)

The expression of mRNA was analyzed in the cells with finished wound healing assays. Total RNA in cells was extracted through TRIzol reagent (Invitrogen, Carlsbad, CA, USA), and complementary DNA (cDNA) was synthesized using 2 μg of total RNA through M-MLV reverse transcriptase (Enzynomics, Daejeon, Republic of Korea) according to the manufacturer’s instructions. The sequences of the primers (Table 1) for human superoxide dismutase type 1 (SOD1), human catalase (CAT), human heme oxygenase 1 (HO-1), human glutathione peroxidase 1 (GPX1), human thioredoxin reductase 1 (TRXR1), human glutathione S-transferase mu 1 (GSTM1), human glutathione S-transferase pi 1 (GSTP1), and human β-actin. PCR was performed for 45 cycles under the following conditions: denaturation at 95 °C for 20 s, annealing at 58 °C for 20 s, and extension at 72 °C for 20 s using a CFX96 (Bio-Rad, CA, USA) with RbTaq™ qPCR 2X PreMIX (Enzynomics, Daejeon, Republic of Korea). The results were normalized to the expression of β-actin.

### 2.8. Experimental Animals

Toxicity tests were performed under the Good Laboratory Practice Regulations for Nonclinical Laboratory Studies (Notification No. 2018-93; Ministry of Food and Drug Safety, Cheongju, Republic of Korea) in accordance with the Test Guidelines for Safety Evaluation of Drugs (Notification No. 2017-71; Ministry of Food and Drug Safety, Cheongju, Republic of Korea). All animal experiments were approved by the Institutional Animal Ethics Committee of Biotoxtech Co., Ltd. (Approval No. 210780; Cheongju, Republic of Korea), which is accredited by the Association for Assessment and Accreditation of Laboratory Animal Care International. For acute toxicity tests, six-week-old male and female Sprague–Dawley (SD) rats were purchased from ORIENT BIO, Inc. (Gyeonggi, Republic of Korea).

For the pharmacokinetic and scratch wound healing study, five-week-old male SD rats were purchased from Samtako Animal, Inc. (Osan, Republic of Korea). All experimental procedures were conducted in accordance with the relevant guidelines for the care of experimental animals and approved by the institutional animal care and use committee (IACUC) of Bioresources and Technology (B&Tech) Co., Ltd., Gwangju, Republic of Korea (Approval No. BT-002-2021). Animals were quarantined before the experiment and adapted to the environment for 1 week. All experimental rats had normal ocular surfaces as observed under a stereomicroscope.

### 2.9. Pharmacokinetic Analysis of PJE

The SD rats used for pharmacokinetic analysis were fasted overnight (16 h) and received a single administration of PJE (5000 mg/kg or 200 mg/kg). Blood samples were collected from the jugular vein at designated time points (0, 0.25, 0.5, 0.75, 1, 2, 3, 6, and 24 h) into ethylenediaminetetraacetic acid (EDTA)-containing tubes. Plasma was harvested by centrifugation at 3000× *g* for 15 min. A total of 30 μL of rat plasma with 10 μL of internal standard (IS) (1000 ng/mL chlorogenic acid-^13^C_3_, S.T.able Inc., Daejeon, Republic of Korea) and 90 μL of extraction solution (methanol:formic acid (100:0.1, *v*/*v*) were added to the test tube, which was vortex-mixed for 3 min and centrifuged at 13,500× *g* for 3 min at 4 °C. Then, each extract was transferred to a test tube, and 60 μL of dilution solution (water:methanol:formic acid (500:500:5, *v*/*v*/*v*) was added before vortexing for 3 min and centrifugation at 13,500× *g* for 3 min at 4 °C. Finally, a 10 μL aliquot was injected into the triple quadrupole liquid chromatography–tandem mass spectrometry (LC–MS/MS) instrument (LCMS-8060, Shimadzu Corp., Kyoto, Japan) equipped with a Shimadzu Nexera UPLC (Shimadzu Corp., Kyoto, Japan) system for analysis. Chromatographic separation was achieved using a ZORBAX SB-C18 column (50 mm × 4.6 mm, 1.8 µm). The nebulizing and drying gas flows were 3.0 L/min and 3.0 L/min, respectively. The pressure of the collision-induced dissociation (CID) gas was 270 kPa. The interface, desolvation line (DL), and heat block temperature were set at 150 °C, 250 °C, and 400 °C, respectively. Mobile phase A was composed of aqueous formic acid (100/0.5, *v*/*v*), and mobile phase B was composed of ACN/formic acid (100/0.5, *v*/*v*). The mobile phase gradient elution program was initially set to 15% mobile phase B, and the composition of mobile phase B increased to 90% (2.01 min), and this concentration was held for 1.5 min, after which the content of mobile phase B decreased to the initial 15% and the system was re-equilibrated for 1.5 min. The flow rate was set to 0.45 mL/min, and the column temperature was maintained at 50 °C. Data were acquired in selected reaction monitoring (SRM) modes with electrospray ionization (ESI).

### 2.10. Evaluation of the Acute Toxicity Induced by a Single Oral Dose of PJE

After the acclimatization period, 20 rats were assigned to two groups (one control (0 mg/kg PJE) and one treatment group (5000 mg/kg PJE); each group contained five male and five female rats) based on their average body weights (males: 171–173 g, females: 134–136 g). Rats received 5000 mg/kg PJE or saline orally once and were observed daily, and the body weights, clinical signs of toxicity, and mortality were recorded for 15 days. Body weights were measured before PJE administration and on days 1, 2, 4, 8, and 15 after administration. On day 15, the surviving rats were sacrificed, and abnormalities in the organs were assessed.

### 2.11. Animal Grouping and Dosing

Rats were randomly divided into six groups (*n* = 10/group) as follows: normal group (NL, normal OS (oculus sinister, left eye); NR, wounded OD (oculus dexter, right eye)), blue light-irradiated group (BL), PJE 25 mg/kg treated group (25), PJE 50 mg/kg treated group (50), PJE 100 mg/kg treated group (100), and PJE 200 mg/kg treated group (200). The OS of each rat in the normal group was used as a normal eye (NL), while each OD was used as the wound control (NR).

All rats in the PJE groups orally received the appropriate dose of PJE once a day for 5 days, and then, corneal wounds were generated. The rat corneal abrasion model was generated as previously reported [31]. Rats were allowed to rest in a dark room for 2 h to recover from anesthesia and irradiated with blue light at 5000 lux for 48 h. The healed area was stained with a fluorescein solution and measured at 0, 12, 24, 36, and 48 h, and rats were allowed to rest for 2 h in a dark room at this time. After the last measurement, the animals were sacrificed, and the eyeballs and plasma were collected for further analysis. Fluorescein staining and tear volume were analyzed as previously described [33]. Histological analysis of cornea such as hematoxylin and eosin (H&E) staining, terminal deoxynucleotidyl transferase dUTP nick end labeling (TUNEL) staining and immunohistochemical staining against interleukin-6 (IL-6) were performed as previously reported [31].

### 2.12. Statistical Analysis

Data are presented as the mean ± standard deviation (SD). The data were statistically evaluated using Student’s *t*-test or two-way analysis of variance (ANOVA) with GraphPad Prism 5 version 5.01 for Windows (GraphPad, Inc., San Diego, CA, USA) software. A value of *p* < 0.05 was considered to indicate statistical significance.

## 3. Results

### 3.1. HPLC Analysis of PJE

HPLC analysis was performed to characterize and identify the natural compounds in PJE and standardize the extracts. The main compounds detected in PJE were CA and NCA and CCA among the chlorogenic acid isomers. The concentrations of CA, NCA, and CCA in PJE were 2.30 ± 0.03 mg/g, 2.27 ± 0.03 mg/g, and 2.57 ± 0.04 mg/g, respectively. Representative chromatograms of the CA, NCA, and CCA reference standards and their corresponding peaks in PJE are shown in Appendix A.

### 3.2. PJE Improved the Migration Activity of and ROS Production in HCECs during Blue Light Irradiation

PJE is an herbal extract that was used in our previous study as a medicinal candidate for delayed corneal abrasion recovery after particulate matter exposure [31]. According to known mechanisms and our previous results, PJE was expected to have protective effects against blue light hazards. Our preliminary experiments established that the application of 1200 lux of blue light is appropriate to delay HCEC migration and increase their production of ROS (Figure 2). HCEC migration decreased to 76.8% (*p* < 0.01) after blue light irradiation compared with the migration of nonirradiated normal cells. PJE-treated cells showed improved migratory activities in a dose-dependent manner under blue light exposure, reaching almost 87% (*p* < 0.01) at both 100 and 300 ng/mL (Figure 2A,B). Cell survival was not affected at any of the indicated concentrations of PJE (Figure 2C). Intracellular ROS levels were significantly increased to 133% (*p* < 0.001) by blue light irradiation, but they were significantly reduced (*p* < 0.001) by 100 and 300 ng/mL PJE treatment (Figure 2D).

### 3.3. Pharmacokinetic Analysis of PJE in Rat Plasma and Acute Toxicity Study

The pharmacokinetic study evaluated the content of chlorogenic acid in rat plasma after oral administration of PJE. Standard HPLC peaks of chlorogenic acid were found at approximately 2 min (Figure 3A). Plasma samples from rats orally administered PJE (5000 mg/kg) once or PJE (200 mg/kg) once a day for one week were analyzed by LC–MS/MS (Figure 3B,C). The plasma chlorogenic acid level peaked at approximately 15–30 min and then slowly decreased, reaching almost zero after 24 h. The acute toxicity of 5000 mg/kg PJE was evaluated in rats, and no changes in clinical signs or body weight were observed (Table 2).

### 3.4. PJE Enhanced Wound Healing and Tear Secretion in Model Rats with Delayed Corneal Wound Healing Induced by Blue Light Irradiation

Corneal epithelial wounds were induced in the OD of rats pretreated with PJE, and the wound healing process was detected at 12 h intervals under blue light irradiation. Each dose of PJE was continuously orally administered once a day until sacrifice. Wound healing was complete 48 h after wound generation in the NR group. Healing was delayed by blue light irradiation but improved by PJE treatment (Figure 4A,B). At 48 h, the wounded area recovered by 66% in the BL group, but healing was dose-dependently enhanced by PJE administration, reaching 92.4% in the 200 mg/kg PJE group (*p* < 0.01); even the lowest concentration of PJE in this study (25 mg/kg) showed a significant result of 84.5% (*p* < 0.05) (Figure 4C). The recovery rates at all time points are shown in Table 3.

Many studies have reported evidence that blue light exposure can cause symptoms similar to those of dry eye syndrome and reduced tear volume, one of the key symptoms exacerbating this disease. Therefore, tear secretion was measured to examine whether blue light and PJE can affect this factor under our experimental conditions. Blue light irradiation reduced tear volume in both eyes (*p* < 0.01), and it was dose-dependently improved by PJE treatment with or without corneal wound generation (Figure 4D,E). Preventing tear reduction may be a beneficial therapeutic action of PJE on corneal wound healing.

### 3.5. Effects of PJE on Corneal Histological Changes

The histological changes in the cornea were analyzed by H&E staining. The NR group showed slight corneal thickening, some infiltrating cells, and newly generated epithelium in the central cornea, but worse morphology was observed in the BL group, such as corneal thickening, a large number of infiltrating cells, and a keratinized surface without epithelium regeneration (Figure 5A). The number of infiltrating cells was determined by calculating the nuclear area in the central and lateral stroma, and this value was significantly increased (*p* < 0.001) in the BL group compared with the NR group (Figure 5B,C). The number of infiltrating cells was dose-dependently reduced in the PJE-treated groups. The rates of epithelium regeneration decreased to 62.3% (*p* < 0.001) in the BL group but reached 94.1% (*p* < 0.001) in the 200 mg/kg PJE-treated group, which is similar to the data shown in Figure 4C (Figure 5D). The area of the immune cell-attached endothelial layer also greatly increased (*p* < 0.001) in the BL group and significantly decreased in a dose-dependent manner in the PJE-treated groups (Figure 5E). Therefore, these results demonstrated that pretreatment with and ongoing administration of PJE inhibited inflammatory cell infiltration and enhanced epithelium regeneration, thereby ameliorating blue-light-induced delayed corneal wound healing.

### 3.6. PJE Inhibited Corneal IL-6 Expression Induced by Blue Light

PJE-mediated modulation of the inflammatory response during delayed wound healing induced by blue light was assessed through immunohistochemical staining against IL-6, a key inflammatory factor involved in wound healing. Almost no IL-6-stained regions were found in the NR group, but the other groups irradiated with blue light displayed increases in the areas of these regions, which were mainly localized to the infiltrating cells in the subsuperficial region of the stroma (Figure 6A). The stained areas were determined via image analysis and were found to increase to 36.0% (*p* < 0.001) in the BL group, an effect that was dose-dependently reduced by PJE treatment, reaching 11.5% (*p* < 0.001) in the 200 mg/kg PJE group (Figure 6B).

### 3.7. PJE Inhibited Corneal Apoptosis Induced by Blue Light

Apoptotic cells were rarely detected in the NR group, but they were markedly increased by blue light irradiation and mainly distributed in the subsuperficial region of the stroma, similar to IL-6-positive regions (Figure 7A). The NR group showed a 5.5% TUNEL-positive area, but in the BL group, this area increased to 33.6% (*p* < 0.001). Moreover, the TUNEL-positive area was reduced to 9.6% (*p* < 0.001) by 200 mg/kg PJE treatment (Figure 7B).

### 3.8. PJE Solvent Fractionation and the Effects of the Fractions on HCEC Wound Healing under Blue Light Irradiation

PJE was separated by solvent fractionation to identify the effector molecules and assess their effect on wound healing delayed by blue light. The HPLC chromatograms of each fraction are shown in Appendix A, and the peaks of CA and its isomers (NCA and CCA) were found in the EtOAc, BuOH, and H_2_O fractions; therefore, these fractions were expected to show the protective effects of PJE. Among the tested concentrations of fractions, clear comparisons of all fractions could be made at concentrations of 1 and 3 μg/mL (Figure 8A). Both the EtOAc and BuOH fractions displayed improved effects, as expected, but the water fraction also displayed significant, dose-dependent increases in activity that were better than the results from the other fractions, even though the active component peak in the HPLC of the water fraction was smaller than that in the EtOAc and BuOH fraction chromatograms (Figure 8B). This may be a result of the purity of the fractions or the existence of residual compounds. Intracellular ROS production induced by blue light was further increased by treatment with the hexane and chloroform fractions (Figure 8C,D). However, the EtOAc, BuOH and water fractions significantly reduced ROS generation (Figure 8E–G). The fractions showed almost no effect on cell survival, with the exception of the hexane fraction (Appendix A).

### 3.9. Fractionation of the PJE Water (PJE/W) Fraction by Open Column Chromatography

The water fraction of PJE was separated again with an HP-20 column, and its effect on wound healing was tested using the same method described above. The HPLC chromatograms of each fraction (F0–F5) are shown in Appendix A. The peaks of the chlorogenic acid isomers were mainly contained in F1~F2, although some small residual peaks appeared in F3~F5. As seen by the HPLC peaks, F1 contained most chlorogenic acid isomers and also showed the best wound healing response among these fractions, whereas F5 showed no recovery and even displayed slight suppression of wound healing at 30 μg/mL (Figure 9A–C). ROS production was also reduced after treatment with F1 and F2 but unchanged after administration of the other fractions (Figure 9D–I). Cell survival was not affected by any of these fractions (Appendix A).

### 3.10. Effects of the Major Components of PJE on HCEC Wound Healing and ROS Production under Blue Light Irradiation

Finally, the CA and its isomers (NCA and CCA) contained within PJE were directly examined in wound healing and ROS assays to confirm their protective effects and to determine whether CA, NCA, and CCA have distinct roles or act synergistically with each other. CA, NCA, and CCA were administered to HCECs at different doses individually, in a dual mixture (1:1) and in a triple mixture (1:1:1). All treated cells displayed significantly enhanced wound healing under blue light irradiation, and the mixtures produced better results than the individual components (Figure 10A). However, intracellular ROS generation was markedly reduced (*p* < 0.001) by low doses of CCA, even though other doses of CA and NCA also significantly affected intracellular ROS generation (Figure 10B–D). Although the CA + NCA mixture showed a better effect at a low dose than each single treatment, the CCA-containing mixtures strongly reduced ROS generation (Figure 10E–H). Cell survival was not changed after treatment with CA and its isomers (NCA and CCA), either alone or in combination (Appendix A).

### 3.11. Effects of PJE Extract and Its Major Compounds on the mRNA Expression of Antioxidant Genes in HCECs under Blue Light Irradiation

Analysis of the expression of genes related to ROS generation was also needed to confirm that the results of the ROS assay came from gene regulation by PJE and its key compounds and to determine whether the superior effect of CCA on ROS reduction, when compared to the effects induced by CA and NCA, was a result of modulating gene expression. The antioxidant genes SOD, CAT, GPX1, GSTM1, and GSTP1 were downregulated by blue light irradiation, whereas HO-1 and TRXR1 were upregulated, but their expression was strongly induced by PJE treatment (Figure 11A). Gene expression changes induced by treatment with CA, NCA, and CCA individually and the triple mixture were analyzed. The three CA isomers effectively increased the expression levels of all genes analyzed. Some genes were upregulated slightly more by CA alone, but the triple mixture showed the highest increases in expression levels for all tested genes (Figure 11B–H). Therefore, the antioxidative properties of PJE and its key components may depend on the expression of these genes, but the specific effect of CCA was not fully supported by these results, as shown above, so the related molecular mechanisms should be further studied.

## 4. Discussion

Increasing concern about blue light exposure has led to many researchers reporting its harmful effects. However, some people have suggested that blue light exposure under ordinary conditions is not toxic in most cases, even after watching displays and being exposed to blue light illumination for a long time, because under these conditions, the hazardous limit of blue light exposure is not exceeded [14]. Most related studies have principally reported that the harmful effects of blue light start with an increase in ROS production and the inflammatory response overall in regions of the eyes. The ocular antioxidative defense systems mainly exist in three parts, the tear film, cornea, and aqueous humor, which contain enzymatic and nonenzymatic antioxidants [34]. Antioxidative enzymes present in these three parts are mainly SODs, CAT, GPXs, glutathione reductase (GR), and glucose-6-phosphate dehydrogenase (G6PD). The SOD proportion in the cornea is large, and this enzyme is the only antioxidative enzyme found in the tear film and aqueous humor [35]. Nonenzymatic antioxidants have been reported to be present in these regions, such as ascorbic acid, glutathione, nicotinamide adenine dinucleotide phosphate (NADPH), uric acid, α-tocopherol, retinol, ferritin, and albumin. Therefore, ocular damage caused by blue light exposure in daily life can be minimized through natural antioxidative defense systems, as described above. However, the eye is susceptible to damage by overloading or wounding the antioxidative system when exposed to environmental risks such as air pollution or irradiation or when the eye experiences dryness, metabolic disorders, surgery, corneal abrasion, or dry eye disease [18,21,31,33]. The cornea is a key defensive region of the eye that has an antioxidative system and serves as a mechanical barrier, although it is vulnerable to environmental injury. The cornea can quickly recover from a shallow scratch on its surface by corneal epithelial cell migration and proliferation, but deep wound healing is disrupted by certain risk factors. Blue light also affects wound healing, but few studies have reported these effects [22,36]. Therefore, the effects of blue light on corneal wound healing and its mechanisms were investigated in this study, and a protective reagent was also suggested.

In most previous reports, cells or animals were directly irradiated with a blue LED, but naked LED lamps are not used in daily life because they induce discomfort. In particular, a prior report noted that LED lamps without diffusers showed higher blue-weighted radiance than those with diffusers, which means that LED light without a diffuser is more harmful, even if the total irradiance is the same [37]. Therefore, the cells and animals in this study were irradiated with blue light through an acryl diffuser (Figure 1). Using diffusers has some advantages: (1) they apply the same irradiation to most regions of the test cells or animals, (2) they block the hot air generated from the LED, and (3) examining the effects of blue light through a diffuser is more similar to realistic conditions.

Oxidative stress is a major cause of epithelial dysfunction [7], so intracellular ROS production was assessed in this study. In preliminary experiments, the optimal blue light irradiation (lux) and exposure time were established to detect ROS production without altering cell survival. At the applied illumination, HCEC wound healing was significantly delayed. PJE effectively reduced ROS production and enhanced wound healing in a dose-dependent manner under the tested conditions (Figure 2). Cells were cultured with serum supplement, in contrast to our previous report, because the cell survival was sensitive and decreased and ROS detection was difficult even when the illuminance was very low. However, the cell migration assay was completed more quickly in this study, which supports the idea that these data are not dependent on cell proliferation.

The toxicity and pharmacokinetics of PJE were assessed before performing the main animal experiments, which suggested that this material was safe at high dosages and provided data that one of the key constituents, chlorogenic acid, is well-absorbed and disappears from plasma within 24 h (Figure 3). To evaluate the toxicity of PJE as a pharmaceutical, further studies on various toxicity tests, such as repeated dose toxicity and carcinogenicity studies, are needed.

Wound healing in corneal abrasion rat models was delayed in vitro but recovered after PJE treatment, and the reduced tear volume was also restored by PJE (Figure 4). The development of dry eye is related to cellular ROS generation and apoptosis [7,38,39]. In addition, corneal wound recovery was delayed under dry eye conditions [40]. Therefore, the reduction in tear volume measured in this study may negatively affect delayed corneal wound healing after blue light irradiation.

Histologically, extensive inflammatory cell infiltration, IL-6 expression levels, and the number of apoptotic cells were increased by blue light, and these changes were reduced by PJE administration (Figure 5, Figure 6 and Figure 7). As shown in the NR group, only a few inflammatory cells were necessary for normal wound healing, which played roles in protecting against microbial infection and enhancing the recovery process by secreting growth factors. These cells are mainly neutrophils and macrophages. Neutrophils have been reported to support re-epithelialization after injury and secrete vascular endothelial growth factor-A to enhance corneal nerve regeneration [41,42,43]. Macrophages also support wound healing by secreting growth factors and clearing debris and apoptotic cells [44,45]. However, excessive infiltration of inflammatory cells and accumulation of apoptotic cells induced by blue light are considered to result from oxidative stress and immune responses that exceed the limits of the natural antioxidative system. PJE treatment improved these deteriorative changes, and it is considered that these preventive effects are a result of its powerful antioxidative fundamental mechanism, although importantly, PJE also enhances the migratory activity of corneal epithelial cells. Therefore, experiments to identify the major components of PJE that are responsible for these protective effects were performed using fractions acquired from solvent fractionation. The results of the wound healing assay and ROS assay were compared to determine if these different functions are dependent on different molecules. CA and its isomers (NCA and CCA), which are major components of PJE, were mainly found in the EtOAc, BuOH, and water (H_2_O) fractions (Appendix A). These fractions improved cell migration and suppressed ROS production (Figure 8) without severe alterations to cell survival, with the exception of the hexane fraction (Appendix A). Unexpectedly, treatment with the water fraction had the best effects on migration and in the ROS assay, although the size of the major peak in its chromatogram was smaller than that in the chromatograms of the EtOAc and BuOH fractions. This may be a result of the purity of the fraction, as seen from the HPLC data, as the residual peaks are smaller in the water fraction than in the other fractions.

The water fraction was further fractionated via HP-20 open column chromatography, and F1 showed the best effects in both experiments, as expected based on the HPLC data (Figure 9). These results are sufficient to confirm that the protective effects of PJE are dependent on three compounds, CA, NCA and CCA. The antioxidative functions of PJE and its component CA isomers have been well-reported previously [30,46,47]. Additionally, CA has beneficial effects on metabolic syndrome [48], cardiovascular diseases [49], hepatic steatosis [50], neurodegenerative diseases [51], and cancer [52]. Diabetic wound healing was also improved by CA [53]. Finally, CA, NCA, and CCA were assessed in both experiments (Figure 10). Cell migration was improved by CA, NCA, and CCA individually with no difference among them, but their triple mixture showed the best effect, indicating that CA, NCA, and CCA act synergistically together. ROS production was also reduced by CA, NCA, and CCA, but CCA showed distinctively powerful effects compared with CA and NCA. These data confirm that CA, NCA, and CCA have distinguishing roles in wound healing under blue light irradiation and that their ability to improve cell migration is not dependent on only their antioxidative function.

The expression of key antioxidant-related genes was quantitated by real-time PCR to confirm that PJE, CA, NCA, and CCA can affect ROS-related gene expression and determine whether CA, NCA, and CCA acted differently (Figure 11). PJE, CA, NCA, and CCA all greatly increased the expression levels of all tested genes. Each gene responded slightly differently to CA, NCA, and CCA, but distinct changes were not found after treatment with CCA, even though HO-1 and TRXR1 were slightly upregulated. The effects of CCA on ROS production and gene expression should be further studied. Additionally, detailed analysis of ROS types and sources such as mitochondria is also necessary. Antioxidative genes such as SOD1, CAT, and GPX1 in our results are already well-known as mitochondrial antioxidant enzymes [54], so PJE is expected to reduce mitochondrial ROS, but it will be confirmed also that each CA may differently affect each type of ROS; as shown, they differently affected gene expression.

There are some limitations in this study: (1) no data directly acquired from human tissue or clinical data; (2) not enough mechanistic suggestions such as migrative mechanisms; (3) effects of each CA single molecule in animals were not tested; (4) comparison with another color of LED was not tested. These would be reported in further studies.

## 5. Conclusions

Blue light significantly delayed corneal wound healing in vitro and in vivo via oxidative stress, inflammation, and apoptosis, as summarized in Figure 12. These detrimental effects of blue light were expected, as they have been reported previously, but blue light applied through a diffuser still causes deleterious changes in the cornea. In particular, blue light makes corneal epithelial cells vulnerable to ROS toxicity by downregulating antioxidant gene expression. However, PJE effectively improved the delayed wound healing in vitro and in vivo, not only enhancing cell migration but also reducing ROS production, inflammation, and apoptosis. The major components of PJE were identified as CA, NCA, and CCA. Each CA isomer effectively improved wound healing and ROS generation, and their mixture synergistically enhanced these effects. These results support the recommendation of using PJE itself as a therapeutic agent. The data in this study provide a new therapeutic strategy for corneal wound healing after exposure to environmental risk factors such as blue light.

## Figures and Tables

**Figure 1 antioxidants-12-01171-f001:**
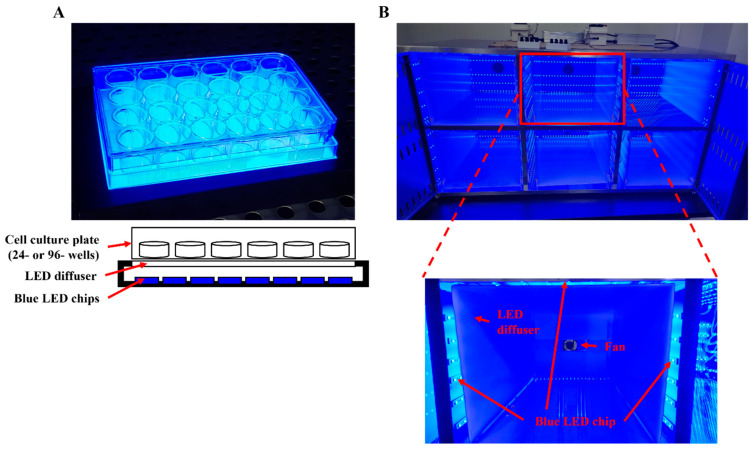
Illuminating devices to apply blue light irradiation to cells and animals. (**A**) Blue backlight for the in vitro experiments. The cell culture plate placed on the panel was irradiated by blue light through an acryl diffuser. (**B**) Animal housing cabinet manufactured with six animal rooms equipped with blue LED strips, a fan, and a light controller. An acrylic diffuser was placed inside each room with an animal cage.

**Figure 2 antioxidants-12-01171-f002:**
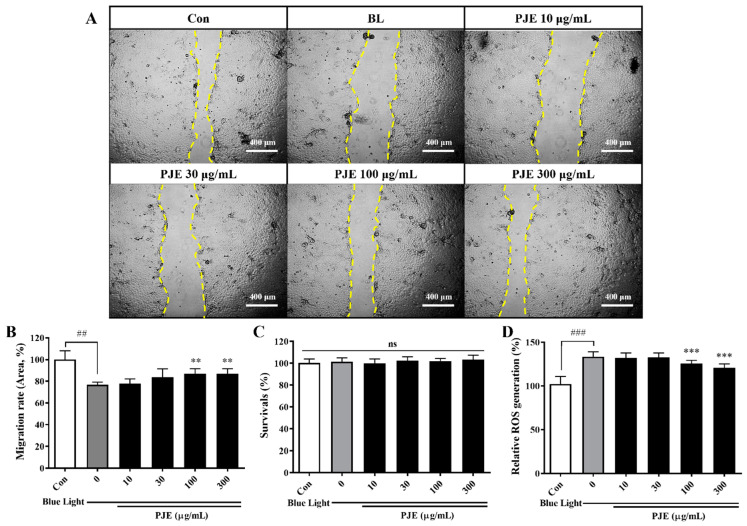
Effects of PJE on wound healing and ROS generation in HCECs under blue light irradiation. (**A**) HCECs treated with PJE for 24 h were scratched. The cells were then irradiated with blue light immediately after measurement of the scratched area (yellow lines). The closed area was measured after 8 h. The scale bars indicate 400 μm. (**B**) The relative migration rates of HCECs expressed as the means ± SDs. (**C**) The survival rates of HCECs are expressed as the means ± SDs. (**D**) Relative intracellular ROS generation after treatment with PJE. The data are presented as the means ± SDs. ^##^
*p* < 0.01, ^###^
*p* < 0.001 compared to Con; ** *p* < 0.01, *** *p* < 0.001 compared to 0 μg/mL PJE.

**Figure 3 antioxidants-12-01171-f003:**
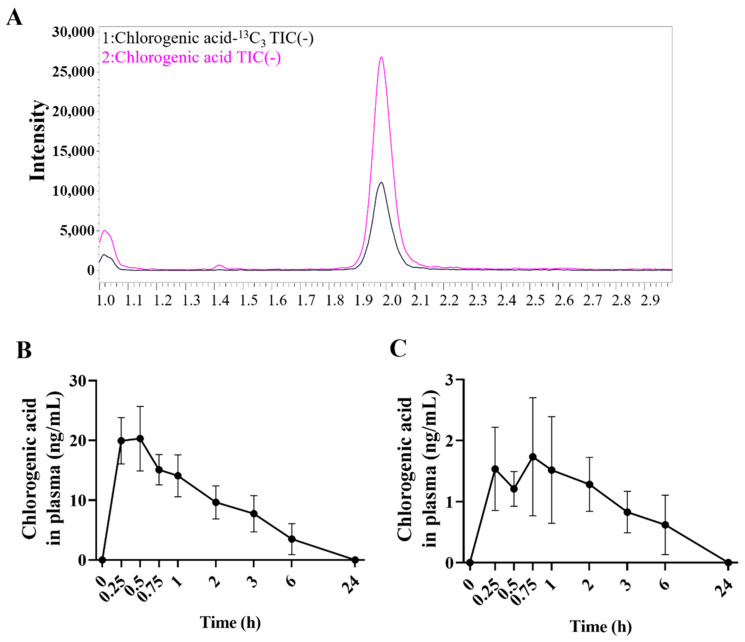
Pharmacokinetic analysis of chlorogenic acid in the plasma of rats after oral PJE administration. (**A**) Standard peaks of chlorogenic acids analyzed by LC–MS/MS. Time course changes in chlorogenic acids in plasma measured immediately after (**B**) a single administration of 5000 mg/kg PJE and (**C**) the last administration of 200 mg/kg PJE (administered once a day for one week).

**Figure 4 antioxidants-12-01171-f004:**
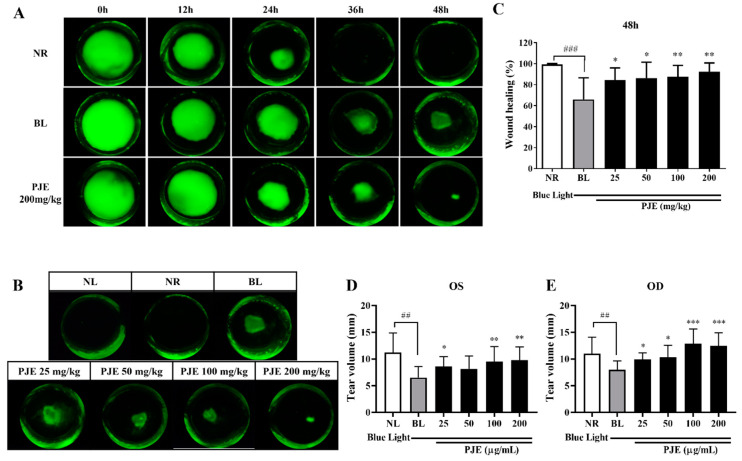
Effect of PJE on rat corneal wound healing under blue light irradiation. Different concentrations of PJE were administered for 5 days, and 4 mm wounds were generated on the right corneas of the rats. The rats were irradiated with blue light during wound healing, and images were obtained at 0, 8, 16, 24, 36, and 48 h with fluorescein staining. (**A**) Corneal fluorescein staining images of the NR, BL, and 200 mg/kg PJE groups during wound healing. (**B**) Representative images of each group at 48 h. (**C**) Relative wound healing areas were calculated and are presented as the means ± SDs. Tear volumes in the (**D**) OS and (**E**) OD are presented as the means ± SDs. ^##^
*p* < 0.01, ^###^
*p* < 0.001 compared to NR; * *p* < 0.05, ** *p* < 0.01, *** *p* < 0.001 compared to BL.

**Figure 5 antioxidants-12-01171-f005:**
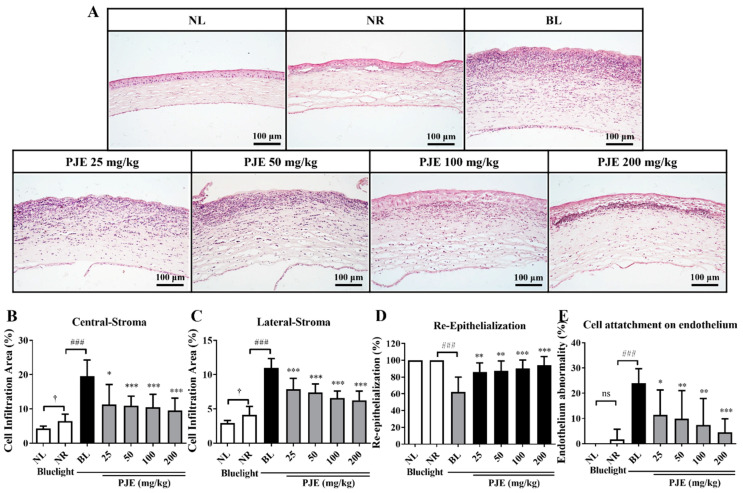
Histological changes during corneal wound healing after treatment with PJE under blue light irradiation. (**A**) Representative hematoxylin and eosin (H&E)-stained images of corneas at 48 h acquired at 200× magnification. Inflammatory cell infiltration in the (**B**) central stroma and (**C**) lateral stroma and (**D**) the re-epithelialization rate and (**E**) cell attachment on endothelium. Data are presented as the means ± SDs. ^†^
*p* < 0.05 compared to NL; ^###^
*p* < 0.001 compared to NR; * *p* < 0.05, ** *p* < 0.01, *** *p* < 0.001 compared to BL; ns, not significant.

**Figure 6 antioxidants-12-01171-f006:**
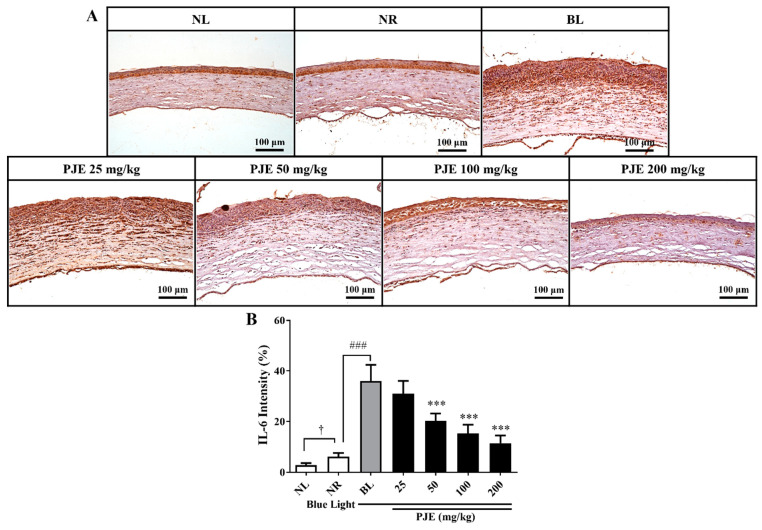
Immunohistochemical analysis of IL-6 expression during corneal wound healing after treatment with PJE under blue light irradiation. (**A**) Representative immunohistochemical staining images for IL-6 in corneas at 48 h acquired at 200× magnification. (**B**) The intensities of the stained areas were calculated, and the data are presented as the means ± SDs. ^†^
*p* < 0.05 compared to NL; ^###^
*p* < 0.001 compared to NR; *** *p* < 0.001 compared to BL.

**Figure 7 antioxidants-12-01171-f007:**
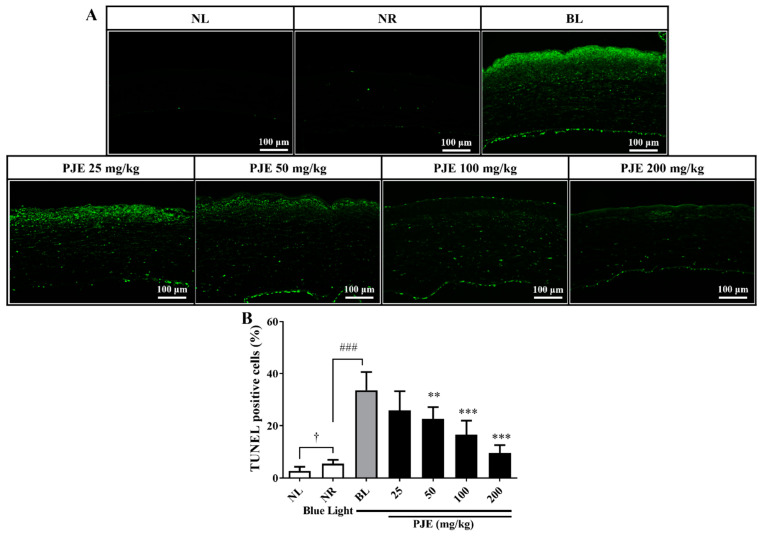
Analysis of apoptotic cells during corneal wound healing after treatment with PJE under blue light irradiation. (**A**) Representative TUNEL staining images of corneas at 48 h acquired at 200× magnification. (**B**) The intensities of the apoptotic cell signals were calculated, and the results are presented as the means ± SDs. ^†^
*p* < 0.05 compared to NL; ^###^
*p* <0.001 compared to NR. ** *p* < 0.01, *** *p* < 0.001 compared to BL.

**Figure 8 antioxidants-12-01171-f008:**
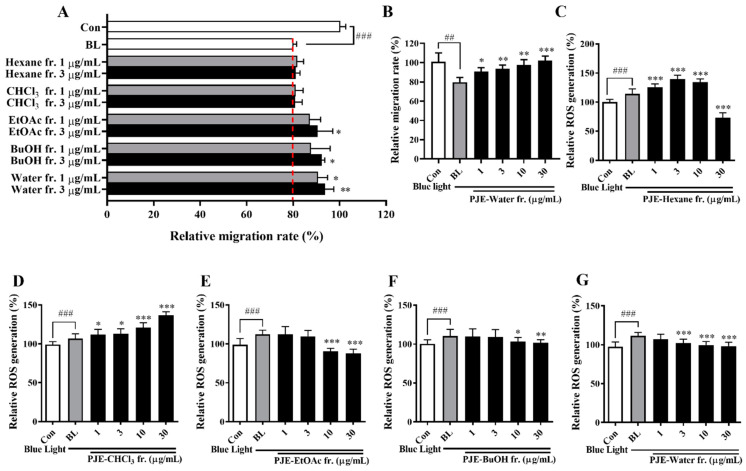
Solvent fractionation of PJE and the effects of these fractions on wound healing and ROS generation in HCECs irradiated with blue light. (**A**) The effect of each fraction on the relative migration rates of HCECs was calculated, and the data from the representative concentrations of 1 μg/mL and 3 μg/mL are presented as the means ± SDs. (**B**) The migration rates of cells treated with the water fraction as representative data are presented for all tested concentrations. Relative intracellular ROS generation after treatment with the (**C**) hexane, (**D**) CHCl_3_, (**E**) EtOAc, (**F**) BuOH, and (**G**) water fractions. Data are presented as the means ± SDs. ^##^
*p* < 0.01, ^###^
*p* < 0.001 compared to Con; * *p* < 0.05, ** *p* < 0.01, *** *p* < 0.001 compared to BL.

**Figure 9 antioxidants-12-01171-f009:**
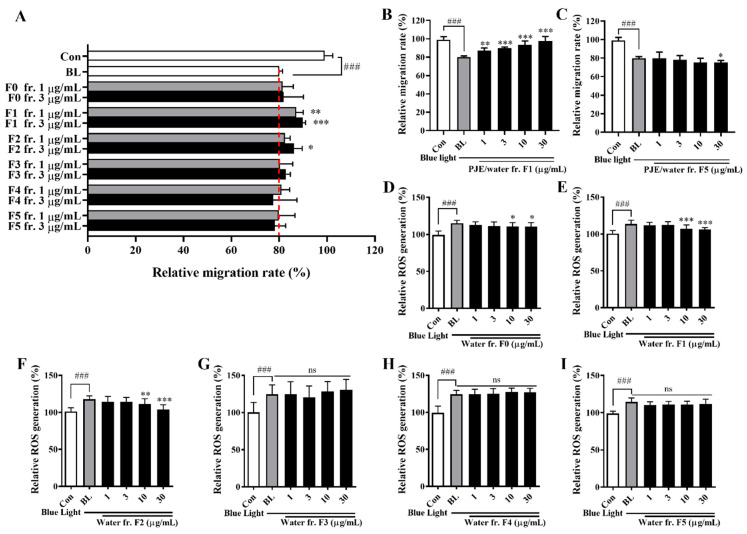
HP-20 open column chromatography of PJE/W and the effects of these fractions on both wound healing and ROS generation in HCECs under blue light exposure. (**A**) The effect of each fraction on the relative migration rates of HCECs was calculated, and the data from the representative concentrations of 1 μg/mL and 3 μg/mL are presented as the means ± SDs. (**B**) Cell migration rates after treatment with (**B**) F1 and (**C**) F5 at all tested concentrations. Relative intracellular ROS generation after treatment with (**D**) F0, (**E**) F1, (**F**) F2, (**G**) F3, (**H**) F4, and (**I**) F5. The data are presented as the means ± SDs. ^###^
*p* < 0.001 compared to Con; * *p* < 0.05, ** *p* < 0.01, *** *p* < 0.001 compared to BL; ns, not significant.

**Figure 10 antioxidants-12-01171-f010:**
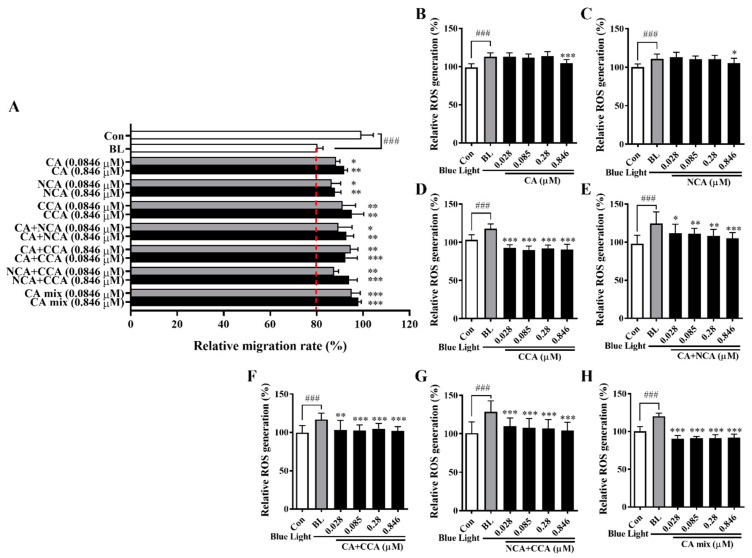
Effects of the major compounds in PJE on the wound healing and ROS generation in HCECs under blue light irradiation. (**A**) The effect of each molecule and their mixtures on the relative migration rates of HCECs was calculated, and the data from the representative concentrations of 0.0846 μM and 0.846 μM are presented as the means ± SDs. Relative intracellular ROS generation after treatment with (**B**) CA, (**C**) NCA, (**D**) CCA, (**E**) CA + NCA, (**F**) CA + CCA, (**G**) NCA + CCA, and (**H**) the mixture of all three CAs. Data are presented as the means ± SDs. ^###^
*p* < 0.001 compared to Con; * *p* < 0.05, ** *p* < 0.01, *** *p* < 0.001 compared to BL.

**Figure 11 antioxidants-12-01171-f011:**
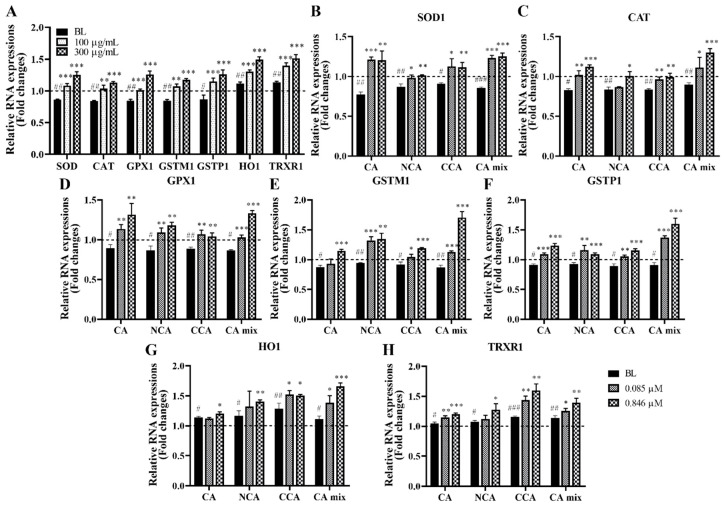
Effects of PJE extract and its major compounds on antioxidant genes in HCECs under blue light exposure. The mRNA expression of antioxidant genes in the cells examined in the scratch wound healing assay. (**A**) Relative changes in the mRNA expression levels of the investigated genes modulated by PJE. The data from the representative concentrations of 100 μg/mL and 300 μg/mL are presented as the means ± SDs. Relative changes in the mRNA levels of (**B**) SOD1, (**C**) CAT, (**D**) GPX1, (**E**) GSTM1, (**F**) GSTP1, (**G**) HO-1, and (**H**) TRXR1 by each molecule and the mixture of the three CAs are presented as the means ± SDs. ^#^
*p* < 0.05, ^##^
*p* < 0.01, ^###^
*p* < 0.001 compared to Con; * *p* < 0.05, ** *p* < 0.01, *** *p* < 0.001 compared to BL.

**Figure 12 antioxidants-12-01171-f012:**
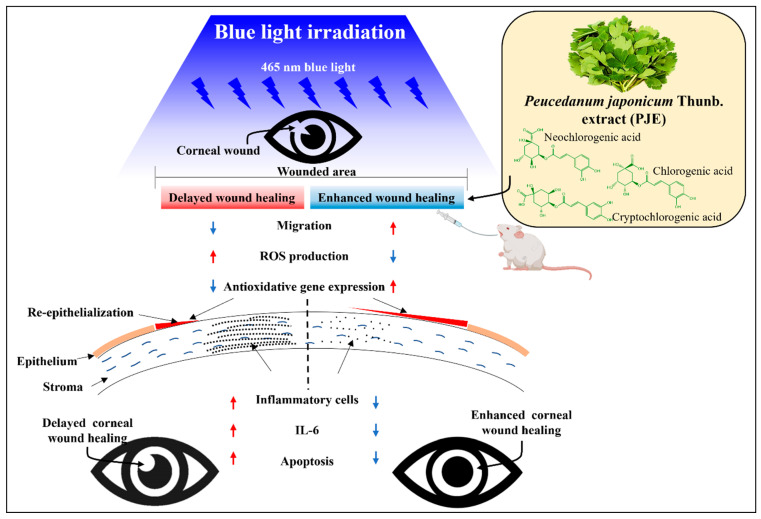
Summary of the effects of PJE on corneal wound healing under blue light irradiation. Oral administration of PJE restored delayed corneal wound healing under blue light irradiation by increasing cell migration and antioxidant gene expression levels, decreasing ROS production in epithelial cells and reducing inflammatory cell infiltration, IL-6 expression levels, and apoptosis in the stromal region of the cornea.

**Table 1 antioxidants-12-01171-t001:** Primer details for real-time PCR analysis.

Gene Symbol	Accession Number	Forward Primer	Reverse Primer
SOD1	NM_000454.5	GGTGGGCCAAAGGATGAAGAG	CCACAAGCCAAACGACTTCC
CAT	NM_001752.4	TGGAGCTGGTAACCCAGTAGG	CCTTTGCCTTGGAGTATTTGGTA
HO-1	NM_002133.3	AAGACTGCGTTCCTGCTCAAC	AAAGCCCTACAGCAACTGTCG
GPX1	NM_000581.4	CAGTCGGTGTATGCCTTCTCG	GAGGGACGCCACATTCTCG
TRXR1	NM_182729.3	ATATGGCAAGAAGGTGATGGTCC	GGGCTTGTCCTAACAAAGCTG
GSTM1	NM_000561.4	TCTGCCCTACTTGATTGATGGG	TCCACACGAATCTTCTCCTCT
GSTP1	NM_000852.4	TTGGGCTCTATGGGAAGGAC	GGGAGATGTATTTGCAGCGGA
β-actin	NM_001101.5	CTCACCCTGAAGTACCCCATC	GGATAGCACAGCCTGGATAGCA

**Table 2 antioxidants-12-01171-t002:** Changes in the body weights of rats exposed to a single dose of PJE.

Sex	Dose	No. of Animals	Parameter	Day 1	Day 2	Day 4	Day 8	Day 15
Male	0 mg/kg	5	Weight	173.1 ± 4.4	189.8 ± 5.3	211.6 ± 5.1	254.1 ± 7.5	322.9 ± 17.7
			Clinical sign	NOA *	NOA *	NOA *	NOA *	NOA *
	5000 mg/kg	5	Weight	171.8 ± 2.3	193.4 ± 4.2	214.7 ± 4.3	256.1 ± 7.6	325.0 ± 18.8
			Clinical sign	NOA *	NOA *	NOA *	NOA *	NOA *
Female	0 mg/kg	5	Weight	135.7 ± 3.5	151.5 ± 2.7	163.7 ± 8.6	180.0 ± 7.4	215.0 ± 7.8
			Clinical sign	NOA *	NOA *	NOA *	NOA *	NOA *
	5000 mg/kg	5	Weight	135.6 ± 5.3	150.1 ± 6.3	161.8 ± 6.0	182.8 ± 8.4	214.7 ± 10.4
			Clinical sign	NOA *	NOA *	NOA *	NOA *	NOA *

The data are presented as the mean ± SD. There were no significant differences. * NOA: no observed abnormality.

**Table 3 antioxidants-12-01171-t003:** Time course of corneal wound healing after PJE treatment under blue light exposure.

		0 h		12 h		24 h		36 h		48 h	
		Average	*p* Value	Average	*p* Value	Average	*p*-Value	Average	*p*-Value	Average	*p*-Value
**PJE** **(mg/kg)**	NR	5.2		20.8		69.6		96.3		99.4	
	BL	2.7	0.066	14.0	0.013 ^#^	35.8	<0.001 ^###^	61.7	<0.001 ^###^	66.0	<0.001 ^###^
	25	3.9	0.456	17.7	0.108	50.0	0.053	69.5	0.251	84.5	0.048 *
	50	1.3	0.197	15.1	0.519	50.5	0.024 *	70.4	0.302	86.2	0.022 *
	100	3.5	0.598	10.3	0.074	43.4	0.189	74.6	0.037 *	87.7	0.003 **
	200	4.6	0.075	10.2	0.051	40.2	0.457	79.2	0.005 **	92.4	0.004 **

The data are presented as the mean ± SD. ^#^
*p* < 0.05, ^###^
*p* < 0.001 compared to NR; * *p* < 0.05, ** *p* < 0.01 compared to BL.

## Data Availability

Data are contained within the article and Appendix A.

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
