# Peer review of "Therapeutic Potential of Peucedanum japonicum Thunb. and Its Active Components in a Delayed Corneal Wound Healing Model Following Blue Light Irradiation-Induced Oxidative Stress"

_antioxidants, 2023, doi:10.3390/antiox12061171_

Round 1

Reviewer 1 Report

Please use designations in Latin for right and left eye in all text as: OD and OS.

Data presented in Figure 2 C and D are data obtained in independent experiments?

 What means it  “The survival rates”? Is cell viability as mentioned in section 2.5. Cell viability?

Table 2 needs to be improved the data presented cannot be read.

Author Response

Responses to Reviewers’ Comments (Reviewer 1)

First of all, thank you for your satisfaction about our manuscript. And according to your comments, we have revised our manuscript carefully.

Q1. Please use designations in Latin for right and left eye in all text as: OD and OS.

(Response) Thank you for your comment. We changed as your suggestion, but the group name, “NL” and “NR” was not changed because it is containing the meaning of “wound generation”. All “right eye” and “left eye” in text were changed.

Q2. Data presented in Figure 2 C and D are data obtained in independent experiments?

(Response) Yes, ROS assay was priorly performed and then, MTT assay was performed in same plate. This result was confirmed again by independent experiment. Cell viability was not different.

Q3. What means it  “The survival rates”? Is cell viability as mentioned in section 2.5. Cell viability?

 (Response) Yes, it is cell viability. Because this result is expressed as percentage of cell viability, usually we expressed like the survival rates. If you think this expression is not proper, it will be changed as “cell viability”.

Q4. Table 2 needs to be improved the data presented cannot be read.

 (Response) OK, it is changed. Thank you.

Reviewer 2 Report

The article „Therapeutic potential of Peucedanum japonicum Thunb. and its active components in a delayed corneal wound healing model following blue light irradiation-induced oxidative stress” by Wan Seok Kang et al. focuses on the effect of blue light on corneal wound healing and its reversal by the leaf extract of Peucedanum japonicum Thunb.

The study uses several in vitro an in vivo-techniques to analyze the effect of blue light on cultivated human corneal cells and on rat corneas. Peucedanum japonicum Thunb. was found to reverse the effects in a dose dependent way.

The experiments were well performed including appropriate controls and including the analysis of possible mechanisms like reactive oxygen species (ROS)-production. The authors used cultivated corneal epithelial cells in proliferation assays, migration assays and detection of apoptosis.

Rats were used for toxicity tests, corneal wound healing tests, measurement of tear volume, apoptosis, inflammatory reactions. The compound was fractionated in high performance liquid chromatography, the components were used in corneal wound healing experiments and confirmed the results. Finally, m-RNA expression related to ROS was studied in cultivated corneal cells.

The authors provide a scientifically well performed study, including several controls and analysis of mechanisms. The article includes microscopic images, histologic photographs, multiple graphs and tables to demonstrate the results.

It would be interesting to transfer the results into a clinical study.

Some improvements in readability could be achieved by organizing the results according to the performed experiments. Table 2 would benefit from a horizontally organized title. The section 2.7 could be organized in a table as it is mainly a summary of the RNA-sequence.

Author Response

Responses to Reviewers’ Comments (Reviewer 2)

Comments and Suggestions for Authors: The article „Therapeutic potential of Peucedanum japonicum Thunb. and its active components in a delayed corneal wound healing model following blue light irradiation-induced oxidative stress” by Wan Seok Kang et al. focuses on the effect of blue light on corneal wound healing and its reversal by the leaf extract of Peucedanum japonicum Thunb.

The study uses several in vitro an in vivo-techniques to analyze the effect of blue light on cultivated human corneal cells and on rat corneas. Peucedanum japonicum Thunb. was found to reverse the effects in a dose dependent way.

The experiments were well performed including appropriate controls and including the analysis of possible mechanisms like reactive oxygen species (ROS)-production. The authors used cultivated corneal epithelial cells in proliferation assays, migration assays and detection of apoptosis.

Rats were used for toxicity tests, corneal wound healing tests, measurement of tear volume, apoptosis, inflammatory reactions. The compound was fractionated in high performance liquid chromatography, the components were used in corneal wound healing experiments and confirmed the results. Finally, m-RNA expression related to ROS was studied in cultivated corneal cells.

 The authors provide a scientifically well performed study, including several controls and analysis of mechanisms. The article includes microscopic images, histologic photographs, multiple graphs and tables to demonstrate the results.

It would be interesting to transfer the results into a clinical study.

 Some improvements in readability could be achieved by organizing the results according to the performed experiments. Table 2 would benefit from a horizontally organized title. The section 2.7 could be organized in a table as it is mainly a summary of the RNA-sequence.

(Response) Thank you for your kind comments. As your comment, primer sequences were expressed as table 1.And Table 2 is changed.

Reviewer 3 Report

In summary, the manuscript contains interesting and potentially important results, however, there are some shortcomings that should be addressed:

1) The measurement of ROS level should be made at different time points. It is suggested to measure it at several time points and pick the most representative time point for mechanistic study.

2) Usually mitochondrial ROS production contributes to cell death. Besides using DCFH-DA, authors have to use mitoSOX to measure mitochondrial ROS production.

3) The authors should add the limits of their work.

Minor editing of English language required

Author Response

Responses to Reviewers’ Comments (Reviewer 3)

Comments and Suggestions for Authors: In summary, the manuscript contains interesting and potentially important results, however, there are some shortcomings that should be addressed:

(Response) First of all, thank you for your satisfaction about our manuscript. And according to your comments, we have revised our manuscript carefully.

Q1.  The measurement of ROS level should be made at different time points. It is suggested to measure it at several time points and pick the most representative time point for mechanistic study.

(Response) We agree with your suggestion. In our preliminary experiment, optimal conditions such as blue light intensity and exposure times for intracellular ROS detection was established. The peak level of ROS was detected at 30 min after 1,200 lux of blue light exposure and then, maintained at slightly decreased levels. Therefore, antioxidative function of PJE was measured at 30 min after BL exposure. As results, 24 h pretreatment of PJE clearly reduced ROS production. However, antioxidative gene expressions were analyzed in the cells after scratch wound healing assay because we expected those genes that expressed early time from injury affected continually wound healing process. Finally, we can clearly show antioxidative gene may reducing ROS production during wound healing under BL exposure.

Q2.  Usually mitochondrial ROS production contributes to cell death. Besides using DCFH-DA, authors have to use mitoSOX to measure mitochondrial ROS production.

(Response) We agree with your suggestion. Detecting mitochondrial ROS is important. Some reports already showed that blue light induced mitochondrial ROS level and this study focused on the wound healing, so we thought that detecting intracellular total ROS levels affecting wide ranges of cellular function is necessary. It is well known that antioxidative enzymes such as SOD, CAT, GPX removed mitochondrial ROS, so our results indirectly proved mitochondrial ROS would be reduced by PJE. In next study, therapeutic mechanisms of PJE will be more deeply explored and not only mitochondrial ROS but also NO (DAF) and extra-mitochondrial superoxide (DHE) be assessed as your suggestion.

We added at 639 lines, “Additionally, detailed analysis of ROS types and sources such as mitochondria also necessary. Antioxidative genes such as SOD1, CAT, GPX1 in our results already well known as mitochondrial antioxidant enzymes [54], so PJE expected reducing mitochondrial ROS, but it will be confirmed also that each CAs may differently affecting each type of ROS as shown they affected differently on gene expression.”

Q3. The authors should add the limits of their work.

(Response) Thank you for your suggestion. We added at 645 lines, “There are some limitations exists in this study. 1) No data directly acquired from human tissue or clinical data. 2) Not enough mechanistic suggestions such as migrative mechanisms. 3) Effects of each CA single molecules in animals were not tested. 4) Comparison with another color LEDs was not tested. These would be reported in further studies.”

Round 2

Reviewer 3 Report

None

Minor editing of English language required